# Short communication: The Topographic Analysis Kit (TAK) for TopoToolbox

Adam M. Forte[1] and Kelin X. Whipple[2]

[1]Department of Geology and Geophysics, Louisiana State University, Baton Rouge, LA
[2]School of Earth and Space Exploration, Arizona State University, Tempe, AZ

**Correspondence:** Adam M. Forte (aforte8@lsu.edu)

**Abstract.** Quantitative analysis of digital topographic data is an increasingly important part of many studies in the geosciences. Initially, performing these analyses was a niche endeavor, requiring detailed domain knowledge and programming skills, but increasingly broad, flexible, open source code bases have been developed to increasingly democratize topographic analysis. However, many of these still require specific computing environments and/or moderate levels of knowledge of both the relevant programming language and the correct way to take these fundamental building blocks and conduct an efficient and effective topographic analysis. To partially address this, we have written the Topographic Analysis Kit (TAK) which leverages the power of one of these open code bases, TopoToolbox, to build a series of high-level topographic analysis tools to perform a variety of common topographic analyses. These include generation of maps of normalized channel steepness or $\chi$ and selection and statistical analysis of populations of watersheds. No programming skills or advanced mastery of Matlab is required for effective use of TAK. In addition, to expand the utility of TAK, along with the primary functions, which like the underlying TopoToolbox functions require Matlab and several proprietary toolboxes to run, we provide compiled versions of these functions that use the free Matlab Runtime Environment for users who do not have institutional access to Matlab or all of the required toolboxes.

## 1 Introduction

The efficient, quantitative analysis of digital topographic data is a primary underpinning of modern tectonic geomorphology research (e.g., Kirby and Whipple, 2012; Whittaker, 2012). Initially, there were a limited number of community standard algorithms to analyze topographic data, including the widely used 'Stream Profiler', a hybrid set of functions between ArcGIS and Matlab for analyzing normalized channel steepness ($k_{sn}$) (Wobus et al., 2006). The code landscape has changed significantly in recent years and several relatively complete and distinct sets of analysis tools and libraries now exist for completing an array of complex topographic analyses, e.g. LSD Topo Tools (e.g., Mudd et al., 2014), TopoTools (Perron, 2010), and TopoToolbox (Schwanghart and Kuhn, 2010; Schwanghart and Scherler, 2014), among others. Of these, TopoToolbox is written in Matlab, making it widely accessible, as Matlab is common in many academic environments and is a relatively easy language to learn.

TopoToolbox is also extremely flexible, serving as a broad code base that is populated with a wide array of versatile functions that do much of the heavy lifting of topographic analysis. On the other hand, TopoToolbox contains few 'finished products', i.e. single functions that allow for complex analysis out of the box. This makes TopoToolbox a powerful community resource, but it also means that using the functions included with TopoToolbox effectively requires 1) an understanding of both the Matlab

language and general programming techniques and 2) a thorough understanding of the correct methodology for chaining together multiple building blocks into an analysis tool tailor-made for the application of interest. Most recently, an increasing number of more complex analysis tools have been built using TopoToolbox, e.g. ChiProfiler for analyzing $k_{sn}$ on streams (Gallen and Wegmann, 2017), KZ-Picker for automatic knickpoint detection (Neely et al., 2017), and DivideTools for analyzing drainage divide stability (Forte and Whipple, 2018). Here we present a new body of functions, the Topographic Analysis Kit

(TAK) that is designed to be a relatively complete set of basic topographic analysis tools that includes a variety of common tasks. These include batch processing of stream net maps and continuous grids of $k_{sn}$ and $\chi$ and fitting $k_{sn}$ values to selected stream profiles that largely replicate and improve upon the original Stream Profiler routines. TAK also includes a variety of tools for the selection of portions of stream networks, projection of longitudinal profiles of stream segments, automated processes for selecting, clipping and analyzing catchment averaged quantities, and construction of multi-variate swath profiles.

Here we describe some of the basic functionality of TAK and provide a representative example of the potential utility of the set of functions for selecting and analyzing watersheds in a basin-averaged approach.

## 2 Principles of Design for TAK

The functions included with TAK are designed to leverage the power and broad codebase of TopoToolbox (Schwanghart and Kuhn, 2010; Schwanghart and Scherler, 2014) and with the following principles in mind: 1) limit the required knowledge of

the Matlab language or general programming techniques by users to successfully, quickly, and robustly analyze topographic data, 2) provide an update to the established methodologies for common tasks (e.g. fitting stream profile segments to measure $k_{sn}$) originally introduced with 'Stream Profiler' (Wobus et al., 2006), 3) bundle together functions for producing common products (e.g. producing maps of $\chi$ and $k_{sn}$) with important controls or preprocessing steps necessary for careful analysis of the outputs (e.g. proper treatment of outlet elevations and incomplete channel networks for maps of $\chi$ and $k_{sn}$ respectively),

4) introduce a framework for efficiently partitioning landscapes into series of small non-overlapping watersheds for a 'basin-averaged' style of topographic analysis (e.g., Bookhagen and Strecker, 2012; Forte et al., 2016), and 5) provide compiled versions of these functions so that users who do not have access to Matlab (or all required toolboxes) can use these tools in a simple environment. In the following sections, we briefly describe the differences between the native Matlab and compiled versions, present the broad types of workflows possible with TAK (Figure 1), and then present a simple case study to show the

type of analysis that is simplified with the basin-averaged style of analysis implemented in TAK. We do not discuss functions or underlying algorithms in detail here, but as a supplement (and within the code repository, see Code Availability) we include a detailed user manual that lays out proper usage of these tools and discusses how they work. Additionally, the header of each function lays out its intended purpose, required and optional inputs, and outputs.

# Possible Workflows Using *Topographic Analysis Kit*

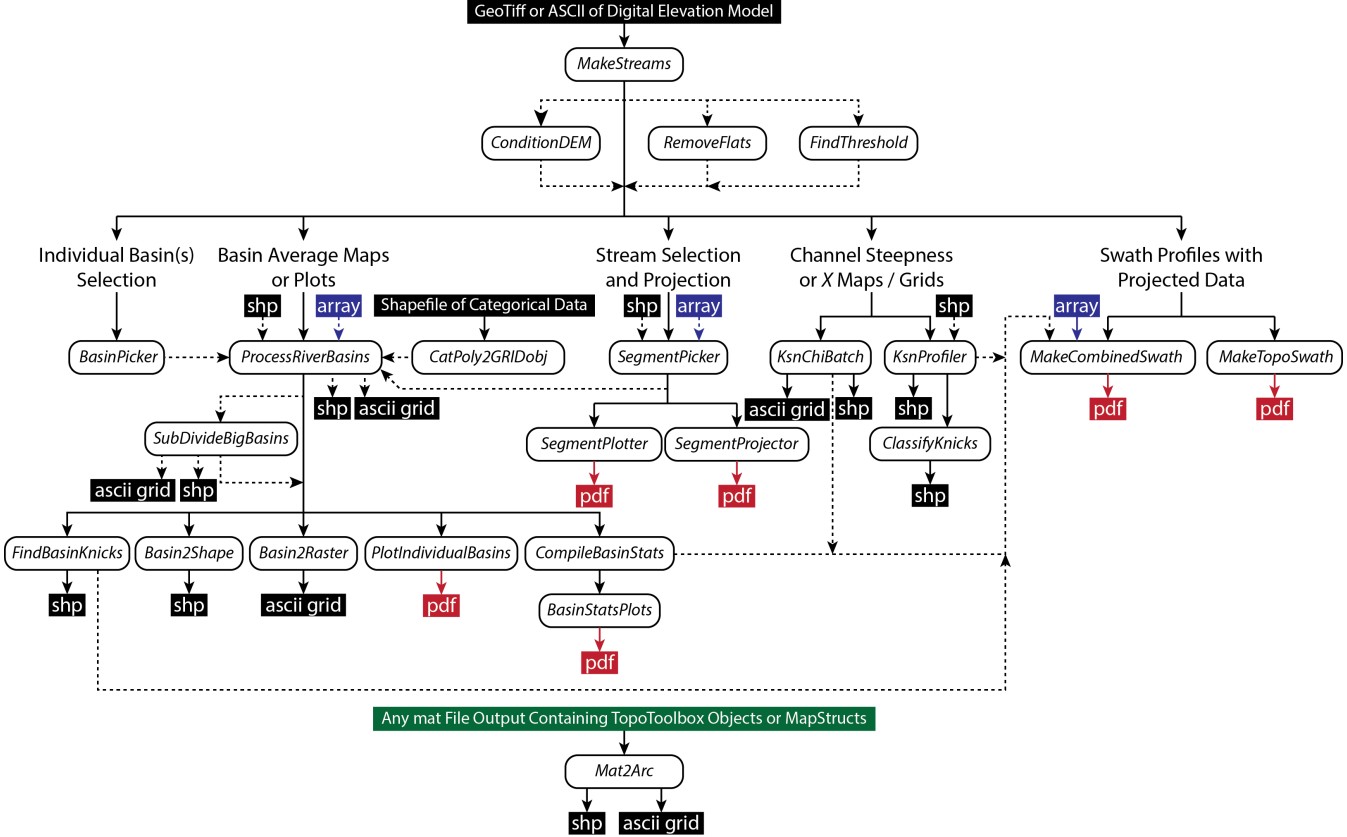

**Figure 1.** Suggested workflows through TAK functions depending on desired outcome and purpose of analysis. Also highlighted are the nature of the outputs produced by different functions. Definitions of inputs and outputs: shp - a shapefile containing vector data, the geometry of which, e.g. points, lines, polygons, depends on the tool in question; array - a Matlab array, i.e. a matrix of numbers; ascii grid - an ESRI Ascii text file that is interpretable as gridded raster data with projection information by many GIS programs; pdf - a figure output as a PDF.

## 3   Matlab vs Compiled Versions

To use the TAK Matlab functions, in addition to having downloaded the free TopoToolbox functions, a user must have licensed versions of Matlab, along with the Image Processing, Mapping, Statistics and Machine Learning, and Optimization Toolboxes, which depending on affiliation could cost upwards of $6150 (prices circa 2018) to fully license. If a user does not have access to Matlab, or all of the required toolboxes, they can instead use the compiled versions. These compiled versions rely on the Matlab Runtime Environment (MRE), which is a free program distributed by Mathworks, the maker of Matlab, for running compiled versions of Matlab code. The GitHub repository (see Code Availability) includes an executable that will download and install the MRE on a user's machine. After the MRE is installed, running the compiled versions is largely similar to running

the Matlab functions, i.e. functions are run from a command line, but instead of being run from a command line within Matlab, they are run from the OS specific command line (terminal in Mac OS X or the command prompt in Windows). Because of limitations on how data can be input into compiled versions of Matlab code, the inputs and outputs differ between some of the Matlab functions and the compiled versions. For example, a function which in the Matlab versions take a two column Matlab array as an input, instead accepts the name of a text file containing a two column array of data (with any standard delimiter) in the compiled versions. Similarly, for TAK Matlab functions which output Matlab proprietary data types (e.g. Matlab arrays or Matlab tables), the compiled versions will instead write out this information as text files or other data types (e.g. ESRI Ascii grids or shapefiles) that are readable by third party programs. A more comprehensive treatment of the different usage of the compiled vs Matlab TAK functions is available in the user manual in the supplement and in the code repository (see Code Availability).

## 4 Possible Workflows

If using TAK exclusively, the entry point for all subsequent functions is the MakeStreams function which generates TopoTool-box versions of the required inputs for subsequent functions, specifically a DEM along with flow routing and stream network information (Figure 1). None of the subsequent functions require use of this initial function. Users may generate valid Topo-Toolbox objects however they see fit, but MakeStreams does offer several built in options for data preparation that may be useful, e.g. automatic identification and removal of true flat areas such as lakes or playas. There are also three companion functions for further basic data preparation for stream profile smoothing (ConditionDEM), removal of mostly flat areas, i.e. those that are not identified with the simple filter in MakeStreams, from stream networks (RemoveFlats), and refinement of stream network definition relating to minimum threshold areas (FindThreshold). Stream smoothing is an essential data preparation step for many topographic analyses and TAK relies on the variety of algorithms included within TopoToolbox to handle smoothing of river profiles (e.g., Schwanghart and Scherler, 2017), all of which are bundled within the ConditionDEM function. As described in the user manual, it is not required that ConditionDEM is run, because by default all TAK functions which require a smoothed river profile will use the 'mincosthydrocon' TopoToolbox function to calculate a linearly interpolated, smoothed channel profile, unless this is overridden by providing an alternatively conditioned DEM produced by the ConditionDEM function. After preparing and/or refining the basic datasets, the pathway through TAK functions depends upon the desired style of analysis or figures, but there are three broad (not mutually exclusive) paths described in the sections below: stream network analysis, basin-averaged analysis, and swath profiles.

### 4.1 Stream Network Analysis

Stream network analysis is a fundamental part of most quantitative topographic investigations and is especially important for tectonic geomorphology. The utility of maps of streams colored by the normalized channel steepness index, $k_{sn}$, for characterizing the active tectonics of erosional landscapes, and specifically using maps of $k_{sn}$ to identify zones of more or less active rates of rock-uplift is well documented (e.g., Kirby and Whipple, 2001; Wobus et al., 2006; Whittaker, 2012; Kirby and Whip-

ple, 2012). Similarly, maps of stream networks colored by $\chi$, as defined by Perron and Royden (2013), are increasingly used to interrogate the topological stability of a stream network (e.g., Willett et al., 2014; Beeson et al., 2017; Forte and Whipple, 2018). In constructing TAK, we have included a variety of functions designed to make stream network analysis simpler. Included within this group of functions are tools for sub-setting stream networks (SegmentPicker), plot selected segments (SegmentPlotter), and projecting portions of longitudinal profiles of streams (SegmentProjector). Also included are tools for generating maps of both $k_{sn}$ and $\chi$ for entire stream networks (KsnChiBatch, e.g. Figure 2B) and for manually fitting $k_{sn}$ values to segments of streams (KsnProfiler). Production of $k_{sn}$ maps with the KsnChiBatch function is largely similar to the results of Stream Profiler, but includes additional methods for aggregating noisy $k_{sn}$ values beyond a simple averaging over a specified length scale, including calculating length averaged $k_{sn}$ values on trunk streams separately from low order streams or calculating length-averaged $k_{sn}$ values on individual stream segments separately (regardless of stream order or size). The production of $\chi$ maps with KsnChiBatch incorporates all of the necessary preprocessing steps described in Forte and Whipple (2018) for ensuring that the $\chi$ values in $\chi$ maps are controlled for outlet elevation and include complete accounting of drainage area. The KsnProfiler function is similar in many ways to the recently published ChiProfiler (Gallen and Wegmann, 2017), but includes some extra functionality modeled after the original Stream Profiler tools (Wobus et al., 2006), e.g., options to manually define the initiation of channels based on slope-area or $\chi$-elevation data and, through the use of the companion ClassifyKnicks function, manually assign classifications to boundaries identified while fitting stream networks. As with the original Stream Profiler, KsnProfiler uses the slope derived from a linear fit of an interpolated version of the $\chi$-elevation relationship to calculate $k_{sn}$ (e.g., Harkins et al., 2007; Perron and Royden, 2013). The primary differences between the original Stream Profiler and KsnProfiler are: 1) use of KsnProfiler does not explicitly require usage of ArcGIS for either picking streams or processing the shapefile (which means it's also significantly faster as the construction of the shapefile in Stream Profiler was the most computationally time-consuming step), 2) users can select segment boundaries on $\chi$-elevation plots in addition to slope-area or longitudinal profiles, 3) there is variety in how streams are selected for analysis including some automated selection schemes, and 4) there is explicit control on how the function deals with overlapping portions of stream networks (i.e. portions of stream networks that could potentially be fit multiple times depending on the streams selected for analysis).

## 4.2 Basin-Averaged Analysis

A common procedure in quantitative topographic analysis is relating topographic metrics (e.g. $k_{sn}$) to an empirical measure of a driving force (e.g., erosion rate) to elucidate more general relationships between surface or tectonic processes and topographic form (e.g., Safran et al., 2005; Cyr and Granger, 2008; Ouimet et al., 2009; DiBiase et al., 2010; Bookhagen and Strecker, 2012; Carretier et al., 2013; Godard et al., 2014; Lague, 2014; Scherler et al., 2014, 2017) or similarly using spatial variations in topographic metrics to infer spatial variation in process or driving forces (e.g., Kirby and Whipple, 2001; Kirby et al., 2003; Hodges et al., 2004; Dorsey and Roering, 2006; Whittaker et al., 2008; Morrell et al., 2015; Adams et al., 2016; Forte et al., 2016; Rossi et al., 2017). In both cases, because of the significant noise inherent in topography, the appropriate way to consider the topographic metric of interest is not strictly on a point or stream section basis, but rather in some spatially averaged form, explicitly in the former (e.g., comparing catchment averaged erosion rates to catchment averaged topographic metrics) and

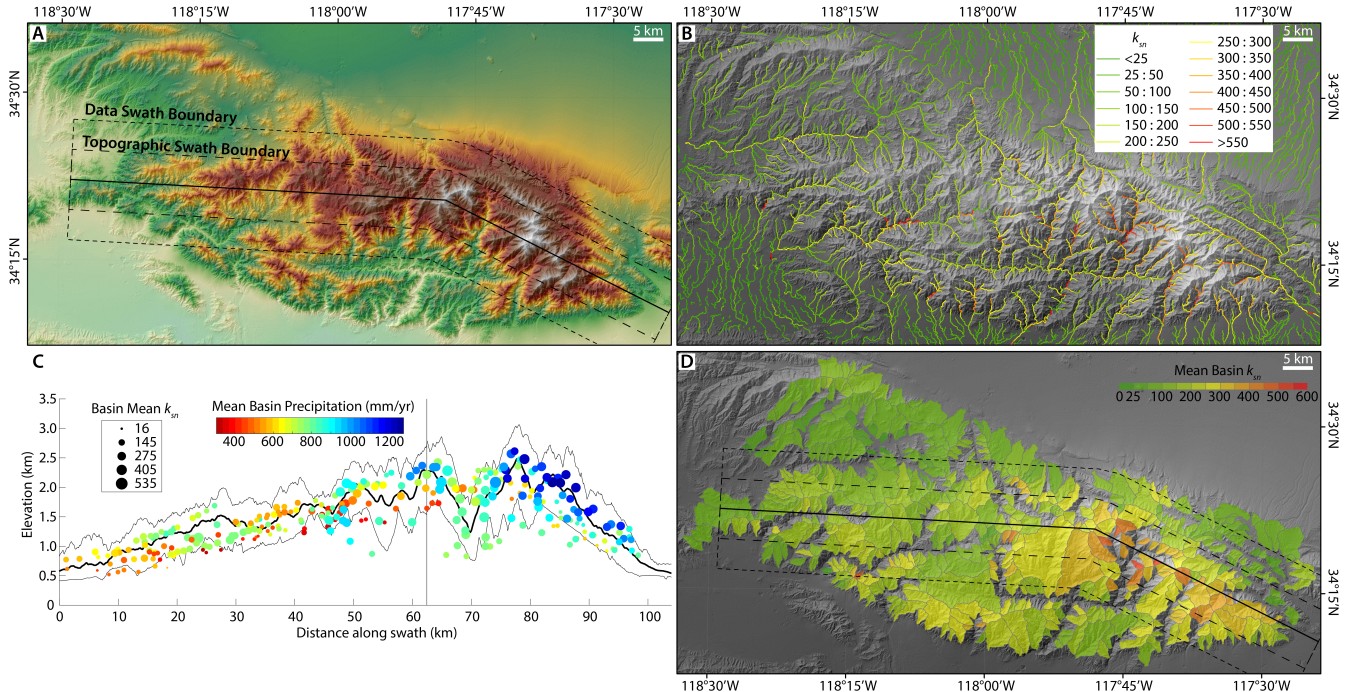

**Figure 2.** Example products output from TAK (with some compilation in ArcGIS and editing in a graphics program). A) Shaded elevation map of the San Gabriel Mountains in southern California with outlines of a combined swath profile. B) Normalized channel steepness map from KsnChiBatch. C) Swath profile with 10 km sampling width for the topography and 20 km sampling width for the basin data, basins are located based on their centroid location and mean elevation, colored by their mean annual precipitation averaged from 1981-2010 (data from PRISM Climate Group, Oregon State University, http://prism.oregonstate.edu, downloaded 1 June 2018), and scaled by their mean $k_{sn}$. D) Map of basin averaged $k_{sn}$ using ProcessRiverBasins and SubDivideBigBasins (using the trunk division method and a max basin size of $25\ \mathrm{km}^2$)

.

more implicitly in the latter. With this idea in mind, it has also been suggested that visualizing and analyzing topographic data (even in the absence of formally spatially averaged empirical quantities like erosion rates) in a basin-averaged sense can be a useful alternative (or complementary) method to traditional stream network analysis for analyzing landscapes (e.g., Bookhagen and Strecker, 2012; Forte et al., 2016). The functions included in TAK for basin-averaged analysis are designed to simplify the creation of maps and plots to analyze data in this way (Figure 2D), making exploratory statistical analysis of spatially averaged topographic data extremely easy. In detail, there is a function that allows for interactive selection of basins to analyze (BasinPicker), the output of which can be directly passed to the main function within this group, ProcessRiverBasins. Selection of basins for ProcessRiverBasins is based on the locations of 'river mouths' (i.e. pour points) and can be selected in TAK using BasinPicker or in a GIS program of the users choice and passed to ProcessRiverBasins as either a list of x, y coordinates or a shapefile of points. Alternatively, ProcessRiverBasins can automatically select basins based on user provided

outlet elevations. ProcessRiverBasins will generate individual Matlab mat files for each watershed containing clipped versions of a variety of grids and vector data (e.g., local relief, maps of $k_{sn}$, etc) including user provided rasters (e.g., precipitation) or polygon shapefiles containing categorical data (e.g., geologic maps) along with statistics for each basin that summarize the clipped basins (e.g., basin averaged local relief, basin averaged $k_{sn}$, etc). There are a variety of companion functions for automatically subdividing these large basins (SubDivideBigBasins, e.g. Figure 2D), manual identification of knickpoints within basins (FindBasinKnicks), plotting profiles of each basin's stream network (PlotIndividualBasin), generating outputs to display these basins as shapefiles (Basin2Shape) or rasters (Basin2Raster), generating compiled Matlab tables of statistics and merging these with other data a user may have for basins, e.g. erosion rates (CompileBasinStats), and basic exploration of relationships between basin averaged values (BasinStatsPlots). To make these functions flexible, but also efficient, the SubDivideBigBasins function can use a variety of criteria to identify pour points within the clipped basins which are then used to 'subdivide' basins (eliminating the need to manually select a large numbers of basin outlets to generate a large population of watersheds). Options for identifying new pour points within larger basins include using the location of confluences, 'outlets' of streams with a particular stream order, and confluences with the trunk stream within a basin network.

### 4.3 Swath Profiles

Swath profiles, which broadly defined are cross-sections through data where that data is sampled across a specified width as opposed to along a single line, can be a useful tool for both analyzing topography but also conveying a more intuitive sense of the topography or other data (e.g., Burbank et al., 2003; Bookhagen and Burbank, 2006). Specifically, swath profiles can be more representative of topography because more traditional cross-sections can be highly biased by the choice of section line location. We include two functions for constructing swath profiles with TAK. The basic MakeTopoSwath is largely a wrapper around the swath construction tool in TopoToolbox but includes additional options to plot the output and directly control the vertical exaggeration of the plots. There is also the MakeCombinedSwath function to create figures pairing topographic swaths with a variety of other point and vector data that is projected onto the swath profile by the function (e.g., Figure 2C). This can be a useful visual tool for displaying a variety of related data in topographic context on a single figure (e.g., Whipple et al., 2016; Forte et al., 2016).

## 5   Case Study of Basin-Averaged Routines

While we feel they are unique in their ease of use or detailed capabilities, the TAK functions related to stream network analysis and swath profiles are largely extensions or incremental improvements of well-established tools and methods, and thus the potential for these functions to be useful is likely self-evident to workers familiar with general principles in topographic analysis. In contrast, the functions related to basin-averaged analyses are largely unique and we believe facilitate a host of possible, large-scale analyses. As an example, we present a simple application of some of these functions to a large-scale problem, specifically exploring broad, potential relationships between topographic form, climate, and rock type. Similar analyses have

been presented before, e.g., Zaprowski et al. (2005), but with substantially smaller datasets, potentially for the simple reason that the absence of tools like the ones we provide makes such analyses incredibly time consuming.

In detail, we use the functions MakeStreams, ProcessRiverBasins, SubDivideBigBasins, CatPoly2GRIDobj, Basin2Shape, and CompileBasinStats along with a SRTM 90 meter digital elevation model of North America, the PRISM mean annual precipitation dataset, and a shape file of compiled state geologic maps from the USGS which contains rock types (Horton et al., 2017) to select and analyze a large suite of watersheds within the continental US (Figure 3). We started the analysis by using the outlets of all streams defined with a threshold area $> 10^9$ km$^2$ as the river mouth input to ProcessRiverBasins. ProcessRiverBasins was run with the result of converting the geologic map shapefile into a raster using the CatPoly2GRIDobj function provided as an optional 'additional categorical grid' and the PRISM dataset provided as an optional 'additional grid'. After initial basin selection, we ran SubDivideBigBasins using the location of confluences with the trunk stream of basins larger than 1000 km$^2$ to automatically subset these basins, which resulted in a total of 1250 individual basins (Figure 3). We then used CompileBasinStats to aggregate all of the data from these basins. From there, because quantities like drainage area, mean $k_{sn}$, mean precipitation, best-fit concavity, percentage of the basin occupied by the most abundant rock type, and the most abundant rock type are automatically calculated by CompileBasinStats, it is simple to filter basins and compare topographic metrics as a function of lithology (Figure 3).

From this point, a variety of observations or comparisons can be made, for example we compare how well this data fits with an expected simplified relationship between $k_{sn}$, uplift rate, and mean precipitation rate, like the one used by D'Arcy and Whittaker (2014),

$$k_{sn} = (U/(K * P^m))^{(1/n)} \tag{1}$$

where U is uplift rate, K is an erosional efficiency constant, P is mean precipitation rate, and m and n are empirical constants (Figure 3). We also explore potential relationships between quantities like basin concavity and mean precipitation, a relationship suggested to exist in some work (e.g., Zaprowski et al., 2005). This at first appears incredibly messy, e.g. the middle panel of plots in Figure 3, but after we begin to further filter the dataset and eliminate basins that may contain large knickpoints (i.e., watersheds without major knickpoints should have nearly linear $\chi$-elevation relationships, and thus the R$^2$ of the $\chi$-elevation relationship, a value included in the results of CompileBasinStats, should be near unity), the range of concavities shrinks dramatically.

Ultimately, we do not wish to interpret too much from these data, but rather we present this as an example of the ease of producing this dataset using TAK. To produce the underlying data for Figure 3 required approximately 5 days of computation time, but less than 1 hour of actual personal interaction with the data. The tasks that required direct interaction with the data were 1) using a GIS program to crop the North America dataset into three large chunks (this was required because of the size of the dataset, the amount of available memory of the desktop computer used to analyze the dataset, and the inherent limitation within Matlab that all data must be loaded into memory), 2) manually filtering the outlets of streams to be included and used by ProcessRiverBasins to avoid streams that primarily drained areas which were not covered by the PRISM or geologic map

datasets, and 3) exploring different criteria of filtering the resulting table of aggregated values. To produce a more robust, interpretable dataset, more time would have been required to more carefully select initial river mouth locations, but the point remains that the functions provided with TAK make such an analysis, incorporating multiple, diverse datasets, extremely easy.

## 6 Conclusions

The functions included within TAK allow a user to quickly and easily perform the majority of 'standard' topographic analyses and especially in the case of the basin-averaged analysis set of functions, expand the scope of the types of analyses which users can perform easily. TAK is built on top of the powerful and flexible TopoToolbox code base and is specifically designed to lower the bar of entry for researchers wishing to include robust, quantitative topographic analysis in their work or teaching, hopefully expanding the community of those using topographic analysis and elevating the quality and reproducibility of published topographic analyses. Additionally, by providing compiled, standalone versions of the TAK functions, we make an effort to expand access of robust and simple topographic analysis to institutions, agencies, organizations, and individuals who do not have access to Matlab, which, while a common fixture in many academic or research settings, is not ubiquitous.

*Code availability.* The TAK functions are available as Matlab code or compiled executables for either Windows or Mac OS X. Matlab functions, executables, and the user manual are available on GitHub (https://github.com/amforte/Topographic-Analysis-Kit). The Matlab functions, executables, and user manual are updated and expanded periodically. The versions of the code referenced in this paper refer specifically to release v.1.0.2. To successfully use the Matlab codes, users must have a licensed copy of Matlab along with licenses for the Mapping Toolbox, Statistics and Machine Learning Toolbox, Optimization Toolbox, and the Image Processing Toolbox. To use the compiled versions, users must install the Matlab Runtime Environment (MRE), which is available for free from Mathworks. For both Mac OS X and Windows, there is a single executable that will install both the MRE and the TAK executable. If users already have MRE installed, the executables are also included as separate files within the GitHub repository. Use of any of these functions in published results should include a reference to this paper.

*Author contributions.* A.M. Forte was responsible for code development and implementation of all TAK functions. K.X. Whipple contributed to theoretical underpinnings of topographic analysis methods, defining desired outputs, and was the primary tester for code resilience. A.M. Forte and K.X. Whipple both contributed to the text.

*Competing interests.* The authors declare that they have no competing interests.

*Acknowledgements.* We thank Nari Miller, Andrew Darling, Joel Leonard, and Wren Raming for beta testing early versions of the TAK functions. We also thank Associate Editor S. Mudd and reviewers A. Duvall, D.E.J. Hobley, and S. Hergarten for helpful suggestions on the original draft of this manuscript. We are indebted to the original authors of TopoToolbox, without which this effort would not have been possible and especially thank Wolfgang Schwanghart for his help with various code issues over the years. Development of TAK was supported by National Science Foundation grant EAR-1450970 to AMF and KXW.

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

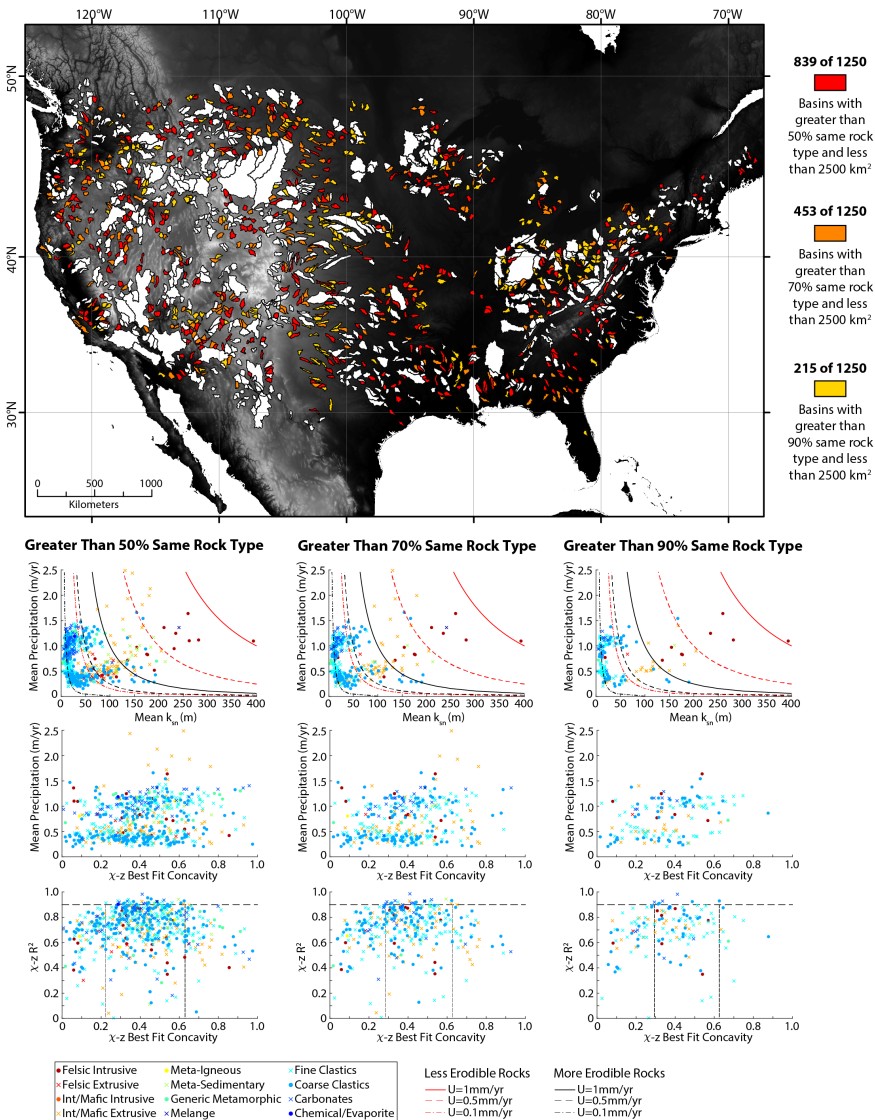

**Figure 3.** Top: Topography of North America showing the extents of the 1250 basins extracted. Colored basins represent basins which meet the established criteria based on drainage area and percent of the catchment occupied by a single rock type. White basins are excluded based on the criteria. Bottom: Plots of basin averaged values comparing the relationships between mean $k_{sn}$ and mean precipitation, best-fit concavity and mean precipitation, and best-fit concavity and the $R^2$ value for the $\chi$-elevation relationship for watersheds with 50% (left), 70% (center), and 90% of the catchment comprised of the same rock type. Colors of the dots indicate the dominant rock type for the watershed. Guidelines on the $k_{sn}$ and mean precipitation plot are calculated using equation (1) and assuming values of m=0.5 and n=1, the range of uplift rates shown at the bottom of the figure, and for two 'K' values equating to less (red) and more (black) erodible rocks. Horizontal line in the bottom plots mark an $R^2$ of 0.90 and the vertical lines indicate the range of concavities above that $R^2$ value. See text for additional description.