# Peer review of "Short communication: The Topographic Analysis Kit (TAK) for TopoToolbox"

_Earth Surface Dynamics, 2018_

## Referee Comment (RC1) · A. Duvall (Referee) · 14 Aug 2018

This short communication, "The Topographic Analysis Kit (TAK) for TopoToolbox" by Forte and Whipple presents a series of streamlined tools and workflows. Building off of the TopoToolbox, these new functions perform a variety of digital topographic data analyses considered essential to many modern tectonic geomorphology studies. As was the case with TopoToolbox and the recent publication of other valuable analysis code such as LSD Topo Tools, the authors provide a big service to the tectonic geomorphology community with this excellent contribution. I expect it will be used heavily by the community.

[Figure]

Because the TAK is designed specifically in the form of workflows to 'finished products' (such as channel steepness maps or topographic swath profiles), and because the authors also released a version of the code to use in the free Matlab Runtime Environment, the toolset holds special promise to break down technologic and economic barriers to quantitative topographic analysis. Beyond its obvious importance as a research tool, I also view the TAK as an important teaching resource, especially because of the detailed Users Guide that accompanies the code and this publication (supplement).

The manuscript is well written, clear and concise, and nicely illustrated. Although I recognize that this is a short communication and is not meant to be a thorough compendium on surface processes, tectonic geomorphology metrics, and fundamentals of landscape analysis, I do think it would benefit from the addition of a few paragraphs before section 3.1. These paragraphs could describe, at least in general, the main value and purpose of the three broad analysis paths (stream network analysis, basin averaged analysis, and swath profiles). The current section 4 (Utility of Basin Averaged Methods) would be bumped up into this section. As is, it seems a little tacked on to the end.

I also suggest a few technical corrections:

Page 1, Line 22: add a comma after accessible and after environments, remove the word and after environments.

Page 2, Line 1: remove the words "perhaps the most" after also

Page 3, Figure 1: Because this toolkit is meant to include even those who are true beginners to digital analysis, you might consider defining some of the terms/abbreviations in this figure in the caption (for example, shp or array).

Page 5, Figure 2: I think panel C should be on the left side and panel D on the right side (swapped as to what it is currently). Usually we read from left to right, so I expected C

to follow B, sitting below A.

Page 5, Line 6: Could add a simple definition of a swath profile at the start of this section.

Page 6, Line 23: Worth noting that beyond academic settings, those without access to Matlab could include local or national government employees and/or consultants who also have a need for this type of topographic analysis tool.

Supplement: TAK Manual v.1.0

Section 3, page 5: ". . ..it is expected that all of the datasets your provide are in the same projection system." "your" should be "you"

Section 8.1, page 22: Update the place holder [Link to Journal Site] after Forte and Whipple (2018) call out.

Section 13, page 56: ". . ..whether submitting and issue" "and" should be "an"

I hope the authors and editors find this review useful. I enjoyed reading the manuscript and working through the TAK code. – Alison Duvall, University of Washington

---

## Referee Comment (RC2) · S. Hergarten (Referee) · 15 Aug 2018

In this short communication a software toolbox for performing morphometric analyses is presented. As quantitative geomorphology apparently becomes more and more important for unraveling the geologic history, the need for software implementations accessible without deeper knowledge in programming has increased in the last years. The toolbox presented here can be seen as some kind of front-end for TopoToolbox and provides more functions "coming out of the box" as well as compiled versions that can be used without MATLAB. I am quite sure that the new toolbox will receive considerable interest although there are already several other toolboxes on different levels available.

[Figure]

However, I am not completely convinced that the type of presentation really fits well into Earth Surface Dynamics. The manuscript itself does not contain much information going beyond listing the main capabilities of the toolbox. In turn the supplement (the manual of the software) is rather technical. When I looking at other examples of papers mainly presenting new tools published in ESurf (e.g., TopoToolbox 2 by W. Schwanghart and D. Scherler, the recently published tools for quantification of changing Earth surface margins by J.M. Lea or my own one on swath profiles) I get the impression that these papers have much more focus on the ideas, algorithms, performance and potential limitations of the presented tools.

For this specific field: We know that river profile analysis can be a challenge in detail, smoothing of river profiles is nontrivial, and the interpretation of $\chi$ maps is also anything but trivial. For a paper (also for a short communication) in ESurf I would expect more information and discussion about the ideas behind the presented toolbox that could indeed make these geomorphic analysis accessible to a larger group of researchers and avoid potential pitfalls. In the present form of the manuscript I would even guess that a note about the availability of the new toolbox and a short presentation of its capabilities on Wolfgang Schwanghart's TopoToolbox web page would do a better job in propagating the software.

---

## Referee Comment (RC3) · D.E.J. Hobley (Referee) · 22 Aug 2018

Review of the Short Communication "The Topographic Analysis Kit (TAK) for TopoToolbox" by Forte and Whipple

In this short submission, the authors present a new front-end interface for the existing TopoToolbox landscape analysis software. The novelty of this software is built largely on enhancing ease of use of the underlying code, on permitting access to the TopoToolbox without a Matlab license, and on enhancing and expanding several of the analyses in TopoToolbox. Given ESurf's recent more explicit focus on landscape evolution software, I would view this submission as appropriate for the journal – especially given its

status as a Short Communication. However, I would happily defer to the AE's view on this. The actual algorithms underlying the analyses are not laid out in the manuscript, though there is an explicit statement of this, and a direction of the reader to the user manual. The material presented feels very much like an incremental advance in terms of science and techniques, but I would say the submission is still warranted in laying out how this front end works, and what advantages it offers the user (a little more on this below).

I believe this submission has no major issues preventing its publication. However, there are in my view a number of moderate niggles where more specific information could be provided to the reader to really sharpen the utility of the text as a description of software functionality. Most of these are listed as technical line items at the end of this text but I wanted to pull out the most significant three beforehand:

1. A large part of the novelty of this submission derives from its use of the Matlab Runtime Environment to avoid the user needing a Matlab licence. However, almost no direct information is provided on this in the body text. This needs rectifying. I would recommend a (short!) new section doing this job either before the current section 3 or after the current section 4. This section should introduce what the MRE is (I've never used it; I imagine that's common); whether this is truly open source (it's not, but that's OK); and what the interface is (graphical? How is it different to "normal" Matlab? Is it web based? Etc etc). You should also state explicitly that the user can deploy TAK in Matlab proper as well. This section should also briefly outline *how* this makes the TAK interface novel – i.e., what are other analysis software packages using as their interfaces, and how is the TAK method better? This kind of detail will let the reader much more easily assess if this is the software package for them. (I appreciate a lot of this is in the Manual, but defending the novelty up front in the actual manuscript seems necessary.) This section should also include a clear and definitive statement on input/output data types – i.e., you say TAK ingests DEMs and shapefiles, but specifically are these ARC formatted only? If not, what else is permitted (NetCDF?

[Figure]

If just naked ASCII, how are projections defined?) For the output, you talk unspecifically in the text about tables and arrays, but you should be specific that these come out as Matlab-proprietary Tables and Arrays, which can be exported to more familiar formats in the MRE (if they can?). Again, I get that this is in the Manual, but it's pertinent up front as it relates to the accessibility of TAK to the user through the Matlab interfaces.

2. The manuscript is explicit that it isn't going to review the underlying algorithms for the analyses. I'm not sure if I'd have made that call, but I think it's OK in the actual manuscript, especially given the Short length. However, I'm not sure that the detail in the Manual is sufficient to cover what's needed either. At the very least, each of the major topographic algorithm sections needs to be anchored to information about the actual methods directly – and ideally, ought to give explicit credit to the designers as well via actual referencing. If these are truly novel techniques either from these authors or novel in TopoToolbox underneath, then say that. Otherwise, you want statements along the lines of "This function implements a modification/implementation of the [whatever reference] algorithm for [doing whatever], as described fully in the Topo-Toolbox manual [direct link to algorithm in the TT manual/webpages]/the TopoToolbox publication". The user shouldn't have to go through the whole chain of (this paper) -> (this manual) -> (TopoToolbox website) -> (TopoToolbox manual index) -> (TopoToolbox section about algorithm) -> (original paper by someone else describing the method) to be able to access this information.

3. Please slightly expand on why the community uses these kinds of tools in the intro, even if just a bit. Probably only a couple of sentences needed, but will anchor the manuscript more strongly – see RC1's comments.

Minor recommendations:

P1Ln 9 – Several sentences in the manuscript get very long and tough to read. Here's one: probably break at "analyses", and restart "These include...".

P2Ln1 – "perhaps the most flexible". This seems like an unnecessary (unjustified?)

judgement call. Either make direct comparisons between TAK and the other options available, or just remove this and stick with being purely descriptive of TAK.

P2Ln12 – again, very long sentence. Break at "…stream profiles. These…"?

P2Ln 21/22 – "with important controls". On what? As written this doesn't make sense to me. Is it just a grammar problem?. Rephrase, and also be more specific.

P2Ln29 – "discusses how they work". I don't think it does; I read this and believed I would find implementation details in the manual, and they aren't there. Probably just delete the clause (or add those algorithms! I'd love this, but I'm guessing you really don't want to…)

Fig. 1 – define "shp", "array" in the caption (see also below)

P4Ln26/27 – This sentence is very vague, both on what the "range of other input types" is, and what the "fully automated procedures" are. Also seems there's a major nonse-quitur here – a procedure is not an input type!? Rewrite for clarity and be more specific about what you mean by both "other input types" and "procedures".

P4Ln28 – "individual files". Again, be specific about which kinds of files (see also 1, above).

P5Ln1 – Ditto. "tables" -> "Matlab Tables".

P5Ln3-5 – You're starting to actually talk methods in the main text here, so in this one specific place I don't think you can get away without an explicit reference to the algorithm(s). i.e., what do you mean by "a variety of schemes", and whose schemes are they?

P6, Code Availability section – A touch more specificity here again, please. I assume those executables you refer to need the Matlab Runtime Environment to work, so say so. Also give information on the fact that a user will need either a licenced Matlab copy, or to download(/use online? I don't know, which is why you should say) the MRE

to make use of the software. Where can they find/download the MRE? Given that the software updates periodically, to comply with code publication best practice please highlight exactly which snapshot of the code this paper specifically describes at time of publication (i.e., as of right now, I would recommend "...expanded periodically; this text refers specifically to the code as of commit 8fad562 on 9th July 2018.") Although you say it in the paper, also add for maximum clarity "The TAK manual is also available through Github, or can be downloaded as a supplement of this publication" (or equivalent statement).

---

## Author Comment (AC1) · 24 Nov 2018

We thank the reviewers for their comments on our manuscript, 'Short Communication: The Topographic Analysis Kit (TAK) for TopoToolbox'. Below we include the original comments of the three reviewers of our manuscript, along with our general response to particular comments (e.g. addressed, rebuttal, etc) followed by a more detailed response describing the action taken to address the comment, or in the case of rebuttals, our reasoning for not specifically addressing the comment.

**Reviewer 1 (A. Duvall)**

[Figure]

**0.0.1**

This short communication, "The Topographic Analysis Kit (TAK) for TopoToolbox" by Forte and Whipple presents a series of streamlined tools and workflows. Building off of the TopoToolbox, these new functions perform a variety of digital topographic data analyses considered essential to many modern tectonic geomorphology studies. As was the case with TopoToolbox and the recent publication of other valuable analysis code such as LSD Topo Tools, the authors provide a big service to the tectonic geomorphology community with this excellent contribution. I expect it will be used heavily by the community.

**0.0.2**

Because the TAK is designed specifically in the form of workflows to 'finished products' (such as channel steepness maps or topographic swath profiles), and because the authors also released a version of the code to use in the free Matlab Runtime Environment, the toolset holds special promise to break down technologic and economic barriers to quantitative topographic analysis. Beyond its obvious importance as a research tool, I also view the TAK as an important teaching resource, especially because of the detailed Users Guide that accompanies the code and this publication (supplement).

**0.0.3**

The manuscript is well written, clear and concise, and nicely illustrated. Although I recognize that this is a short communication and is not meant to be a thorough compendium on surface processes, tectonic geomorphology metrics, and fundamentals of landscape analysis, I do think it would benefit from the addition of a few paragraphs

before section 3.1. These paragraphs could describe, at least in general, the main value and purpose of the three broad analysis paths (stream network analysis, basin averaged analysis, and swath profiles). The current section 4 (Utility of Basin Averaged Methods) would be bumped up into this section. As is, it seems a little tacked on to the end.

0.0.4   Reply:

*Addressed:* We agree that providing a little more context for the utility of these techniques would be good. Instead of doing exactly what was suggested by Reviewer 1 (and 3), we have included these descriptions at the beginning of each subsection (i.e. Stream Network Analysis, Basin Averaged Analysis, and Swath Profiles). The text discussing the utility of the Basin Averaged Analysis (originally Section 4) has been moved to the front of the Basin Averaged Analysis tools section and a new section (with a new, more substantial example) has been added.

0.0.5

I also suggest a few technical corrections:

0.0.6

Page 1, Line 22: add a comma after accessible and after environments, remove the word and after environments.

0.0.7   Reply:

*Addressed:* Change made.
**0.0.8**

Page 2, Line 1: remove the words "perhaps the most" after also

**0.0.9  Reply:**

*Addressed:* Change made.

**0.0.10**

Page 3, Figure 1: Because this toolkit is meant to include even those who are true be-
ginners to digital analysis, you might consider defining some of the terms/abbreviations
in this figure in the caption (for example, shp or array).

**0.0.11  Reply:**

*Addressed:* We have defined, or clarified, what the input and output data types mean
in this figure within the caption.

**0.0.12**

Page 5, Figure 2: I think panel C should be on the left side and panel D on the right side
(swapped as to what it is currently). Usually we read from left to right, so I expected C
to follow B, sitting below A.

0.0.13   Reply:

*Addressed:* We have made the suggested change.

0.0.14

Page 5, Line 6: Could add a simple definition of a swath profile at the start of this section.

0.0.15   Reply:

*Addressed:* We have added a simple definition of a swath profile in the opening line of this paragraph.

0.0.16

Page 6, Line 23: Worth noting that beyond academic settings, those without access to Matlab could include local or national government employees and/or consultants who also have a need for this type of topographic analysis tool.

0.0.17   Reply:

*Addressed:* We have expanded the list of end users who might find the compiled versions useful.

0.0.18

Comments on Supplement: TAK Manual v.1.0

0.0.19

Section 3, page 5: ". . ..it is expected that all of the datasets your provide are in the same projection system." "your" should be "you"

0.0.20   Reply:

*Addressed:* The error has been fixed.

0.0.21

Section 8.1, page 22: Update the place holder [Link to Journal Site] after Forte and Whipple (2018) call out.

0.0.22   Reply:

*No Change Made:* The text in question is an actual hotlink to the journal site including the PDF of the referenced paper.

0.0.23

Section 13, page 56: ". . ..whether submitting and issue" "and" should be "an"
0.0.24    Reply:

*Addressed:* The error has been fixed.

0.0.25

I hope the authors and editors find this review useful. I enjoyed reading the manuscript and working through the TAK code. – Alison Duvall, University of Washington

**Reviewer 2 (S. Hergarten)**

0.0.26

In this short communication a software toolbox for performing morphometric analyses is presented.  As quantitative geomorphology apparently becomes more and more important for unraveling the geologic history, the need for software implementations accessible without deeper knowledge in programming has increased in the last years. The toolbox presented here can be seen as some kind of front-end for TopoToolbox and provides more functions "coming out of the box" as well as compiled versions that can be used without MATLAB. I am quite sure that the new toolbox will receive considerable interest although there are already several other toolboxes on different levels available.

0.0.27

However, I am not completely convinced that the type of presentation really fits well into Earth Surface Dynamics. The manuscript itself does not contain much information going beyond listing the main capabilities of the toolbox.  In turn the supplement (the manual of the software) is rather technical. When I looking at other examples of papers

mainly presenting new tools published in ESurf (e.g., TopoToolbox 2 by W. Schwang-hart and D. Scherler, the recently published tools for quantification of changing Earth surface margins by J.M. Lea or my own one on swath profiles) I get the impression that these papers have much more focus on the ideas, algorithms, performance and potential limitations of the presented tools.

0.0.28   Reply:

*Rebuttal:* While we concede that the Short Communications highlighted by the reviewer have focused on algorithms or limitations, we would argue that the content of our manuscript, at least with our understanding of the guidelines, fits within the description of this manuscript type, "Short communications report new developments, significant advances, and novel aspects of experimental, modelling, and theoretical methods and techniques which are relevant for scientific investigations within the journal scope. Manuscripts of this type should be short (a few pages only). Highly detailed and specific technical information such as computer programme code or user manuals can be included as electronic supplements." Ultimately, whether the manuscript in it's current, revised, form fits within the definition of a Short Communication paper will be up to the editor.

0.0.29

For this specific field: We know that river profile analysis can be a challenge in detail, smoothing of river profiles is nontrivial, and the interpretation of $\chi$ maps is also anything but trivial. For a paper (also for a short communication) in ESurf I would expect more information and discussion about the ideas behind the presented toolbox that could indeed make these geomorphic analysis accessible to a larger group of researchers and avoid potential pitfalls. In the present form of the manuscript I would even guess

that a note about the availability of the new toolbox and a short presentation of its capabilities on Wolfgang Schwanghart's TopoToolbox web page would do a better job in propagating the software.

**0.0.30 Reply:**

*Partially Addressed, Primarily Rebuttal:* While we appreciate the reviewer's opinion, there is little in terms of actionable changes that can be made based on this comment. The reviewer highlights both profile smoothing and the calculation of chi as areas where better documentation is required, however in both the main text and supplement we are explicit that we are using the smoothing algorithms implemented in TopoToolbox (and cite the paper where those algorithms were described). Similarly, in the original submission we describe in the supplement that the functions for calculating chi maps are based on our previously published work (Forte Whipple, 2018, EPSL) that goes through in detail the considerations necessary for constructing a chi map (we have added a brief description and citation to this in the main text in the revised version). In regards to whether 'a note about the availability of the new toolbox and a short presentation of its capabilities on Wolfgang Schwanghart's TopoToolbox web page would do a better job in propagating the software', this may be true. This is a great suggestion for increasing visibility of our paper. We hope to be able to add a blog post about TAK to Wolfgang's web page that includes a link to a published esurf Short Communication. However, we still see great value in primarily publishing and disseminating this work through the esurf Short Communications venue.

**Reviewer 3 (D.E.J. Hobley)**

0.0.31

In this short submission, the authors present a new front-end interface for the existing TopoToolbox landscape analysis software. The novelty of this software is built largely on enhancing ease of use of the underlying code, on permitting access to the TopoToolbox without a Matlab license, and on enhancing and expanding several of the analyses in TopoToolbox. Given ESurf's recent more explicit focus on landscape evolution software, I would view this submission as appropriate for the journal – especially given its status as a Short Communication. However, I would happily defer to the AE's view on this. The actual algorithms underlying the analyses are not laid out in the manuscript, though there is an explicit statement of this, and a direction of the reader to the user manual. The material presented feels very much like an incremental advance in terms of science and techniques, but I would say the submission is still warranted in laying out how this front end works, and what advantages it offers the user (a little more on this below).

0.0.32 Reply:

*Rebuttal:* There is indeed no intended significant advance in either science or underlying algorithmic technique (generally speaking). This is not hidden – as all reviewers have recognized, we are describing and disseminating a powerful, easy to use, and open-access, open-source "front end" to the powerful TopoToolbox which is otherwise limited by the substantial learning curve required to implement desired analyses. So ultimately there are few clever new algorithms to be described. Rather than accomplishing a scientific or fundamental algorithmic technical advance, our aim is to facilitate potential scientific advances by the readership of esurf. As this apparent misunderstanding dominates this reviewer's comments, we have clarified this point in both the revised text and revised user manual: all underlying functions not explicitly de-
scribed in the user manual are contained within Matlab or TopoToolbox and have been described elsewhere.

0.0.33

I believe this submission has no major issues preventing its publication. However, there are in my view a number of moderate niggles where more specific information could be provided to the reader to really sharpen the utility of the text as a description of software functionality. Most of these are listed as technical line items at the end of this text but I wanted to pull out the most significant three beforehand:

0.0.34

1. A large part of the novelty of this submission derives from its use of the Matlab Runtime Environment to avoid the user needing a Matlab licence. However, almost no direct information is provided on this in the body text. This needs rectifying. I would recommend a (short!) new section doing this job either before the current section 3 or after the current section 4. This section should introduce what the MRE is (I've never used it; I imagine that's common); whether this is truly open source (it's not, but that's OK); and what the interface is (graphical? How is it different to "normal" Matlab? Is it web based? Etc etc). You should also state explicitly that the user can deploy TAK in Matlab proper as well. This section should also briefly outline *how* this makes the TAK interface novel – i.e., what are other analysis software packages using as their interfaces, and how is the TAK method better? This kind of detail will let the reader much more easily assess if this is the software package for them. (I appreciate a lot of this is in the Manual, but defending the novelty up front in the actual manuscript seems necessary.) This section should also include a clear and definitive statement on input/output data types – i.e., you say TAK ingests DEMs and shapefiles,

but specifically are these ARC formatted only? If not, what else is permitted (NetCDF? If just naked ASCII, how are projections defined?) For the output, you talk unspecifically in the text about tables and arrays, but you should be specific that these come out as Matlab-proprietary Tables and Arrays, which can be exported to more familiar formats in the MRE (if they can?). Again, I get that this is in the Manual, but it's pertinent up front as it relates to the accessibility of TAK to the user through the Matlab interfaces.

**0.0.35 Reply:**

*Addressed:* We have added a new section 'Matlab vs Compiled Versions' which describes (briefly) the difference between the Matlab and compiled version of TAK and clarifies how the compiled version deals with proprietary Matlab data types, e.g. if the Matlab version takes a Matlab array as input, the compiled version then instead takes a text file that includes the same data or similarly, if the Matlab version outputs a Matlab array, the compiled version will output a text file containing the same data. We have also added some clarification to section 3 of the user manual (Preparing Datasets for TAK) on which file formats are supported for input and output. All of the inputs and outputs (not considering the Matlab functions which output Matlab proprietary data types) are generally file types readable by a variety of programs. E.g. While ESRI ASCII files and shapefiles were initially proprietary data types for raster and shapefiles, respectively, both of these are readable and writeable by most standard open source GIS tools (e.g. QGIS, GDAL, OGR, etc).

**0.0.36**

2. The manuscript is explicit that it isn't going to review the underlying algorithms for the analyses. I'm not sure if I'd have made that call, but I think it's OK in the actual manuscript, especially given the Short length. However, I'm not sure that the detail

in the Manual is sufficient to cover what's needed either. At the very least, each of the major topographic algorithm sections needs to be anchored to information about the actual methods directly – and ideally, ought to give explicit credit to the designers as well via actual referencing. If these are truly novel techniques either from these authors or novel in TopoToolbox underneath, then say that. Otherwise, you want statements along the lines of "This function implements a modification/implementation of the [whatever reference] algorithm for [doing whatever], as described fully in the Topo-Toolbox manual [direct link to algorithm in the TT manual/webpages]/the TopoToolbox publication". The user shouldn't have to go through the whole chain of (this paper) -> (this manual) -> (TopoToolbox website) -> (TopoToolbox manual index) -> (TopoToolbox section about algorithm) -> (original paper by someone else describing the method) to be able to access this information.

0.0.37   Reply:

*Partially Addressed:* We appreciate the reviewer's goal of ensuring that users are provided with an in-depth understanding of what software is doing and of course that prior work is properly attributed. With the original and especially the revised text and user manual we feel that we have achieved this goal. As stated above we now make it very clear that all fundamental underlying algorithms are part of Matlab or TopoToolbox and have been described elsewhere. We do not replace or re-write any core TopoToolbox functions, but rather harness them in a user-friendly front-end, the user operation of which is described in detail in a user manual . for TAK in which we have attempted to document virtually all functionality included in TAK. Interestingly, the user manual elicited opposite reactions from 2 of the 3 reviewers, with reviewer 2 considered it 'rather technical' where as reviewer 3 indicates here that it is not technical enough. An important clarification from above is worth repeating: there are virtually no new underlying algorithms in TAK to be described. All the fundamental algorithms (for extracting a stream network, for example) are within TopoToolbox and have been described

in published papers, e.g. Schwanghart Kuhn, 2010; Schwanghart Scherler, 2017; Schwanghart Scherler 2017. Moreover, the first reviewer in contrast was quite happy with the original manual. Given this context, to partially address this particular comment we have added even more detail on what is being done with various functions and/or particular options of functions throughout the user manual. However, in most cases, there are no new algorithms to be described. For functions which are implementing some sort of published algorithm or procedure, we were already explicit in referencing the prior work (e.g. we already specified that the methodology by which normalized channel steepness was calculated is the same as in Stream Profiler and described in Wobus et al, 2006). We have also generally tried to clarify that in most cases, these codes are not establishing new algorithms or procedures, but rather using standardized procedures. Beyond instances like this, it seems like addressing the reviewers comment would require compiling a list of every Matlab or TopoToolbox function used in each TAK function and spelling that out in the user manual, which seems both excessive and counterproductive. Finally, we do want to highlight again that as described in what constitutes a Short Communication, the guidelines are somewhat explicit that the main submission be short , i.e. 'a few pages only' and that, 'Highly detailed and specific technical information such as computer programme code or user manuals can be included as electronic supplements'.

0.0.38

3. Please slightly expand on why the community uses these kinds of tools in the intro, even if just a bit. Probably only a couple of sentences needed, but will anchor the manuscript more strongly – see RC1's comments.

0.0.39   Reply:

*Addressed:* Good idea, see response to similar comment by Reviewer 1.

0.0.40   Minor recommendations:

0.0.41

P1Ln 9 – Several sentences in the manuscript get very long and tough to read. Here's one: probably break at "analyses", and restart "These include. . ."

0.0.42   Reply:

*Addressed:* We have made this highlighted change and have generally tried to shorten or break up sentences in other places in the manuscript to make it more readable.

0.0.43

P2Ln1 – "perhaps the most flexible". This seems like an unnecessary (unjustified?) judgement call. Either make direct comparisons between TAK and the other options available, or just remove this and stick with being purely descriptive of TAK.

0.0.44   Reply:

*Addressed:* This phrase was removed and now simply indicates that TopoToolbox is flexible.

0.0.45

P2Ln12 – again, very long sentence. Break at ". . .stream profiles. These. . ."?

0.0.46   Reply:

*Addressed:* We have broken this sentence up per the suggestion.

0.0.47

P2Ln 21/22 – "with important controls". On what? As written this doesn't make sense to me. Is it just a grammar problem?. Rephrase, and also be more specific.

0.0.48   Reply:

*Addressed:* We have rephrased this sentence to be more clear that we were trying to bundle the production of particular products, like normalized channel steepness maps, with important pre-processing steps, like ensuring that the stream network has a complete accounting of drainage area for all streams, so that inexperienced users would at least be aware that these are things you should do as part of these analyses.

0.0.49

P2Ln29 – "discusses how they work". I don't think it does; I read this and believed I would find implementation details in the manual, and they aren't there. Probably just delete the clause (or add those algorithms! I'd love this, but I'm guessing you really don't want to. . .)

0.0.50   Reply:

*Partially Addressed:* We are unclear exactly what the reviewer was hoping to find in the manual, but we would dispute the idea that 'implementation details. . . aren't there' as we generally include descriptions of what the codes are doing or how the behavior of the code changes depending on the options being used by the user. As stated above, we now make it clearer that all fundamental underlying functions are contained within Matlab or TopoToolbox. Also note that we are after all also releasing the source code of our "front end". We have attempted to strike a balance between describing all operational details in the manual and overwhelming users with too much detail. That being said, in the revised user manual, we have provided additional details for some options and functions where appropriate. At this point it is rather exhaustive as a user manual and is now nearly 100 pages long.

0.0.51

Fig. 1 – define "shp", "array" in the caption (see also below)

0.0.52   Reply:

*Addressed:* We made this change in response to a similar comment from Reviewer 1.

0.0.53

P4Ln26/27 – This sentence is very vague, both on what the "range of other input types" is, and what the "fully automated procedures" are. Also seems there's a major non-sequitur here – a procedure is not an input type!? Rewrite for clarity and be more specific about what you mean by both "other input types" and "procedures".

0.0.54   Reply:

*Addressed:* We have rewritten this sentence (now two sentences) specifying how users can select basins for the ProcessRiverBasins function.

0.0.55

P4Ln28 – "individual files". Again, be specific about which kinds of files (see also 1, above).

0.0.56   Reply:

*Addressed:* We have specified that these are Matlab mat files.

0.0.57

P5Ln1 – Ditto. "tables" -> "Matlab Tables".

0.0.58   Reply:

*Addressed:* Change made.

0.0.59

P5Ln3-5 – You're starting to actually talk methods in the main text here, so in this one specific place I don't think you can get away without an explicit reference to the

algorithm(s). i.e., what do you mean by "a variety of schemes", and whose schemes are they?

0.0.60   Reply:

*Addressed:* We have modified this sentence to clarify that by 'scheme' we mean criteria for identifying new pour points for smaller basins within larger basins. We have also clarified the following sentence, which did list these criteria in the original draft, but is now more explicit.

0.0.61

P6, Code Availability section – A touch more specificity here again, please. I assume those executables you refer to need the Matlab Runtime Environment to work, so say so. Also give information on the fact that a user will need either a licenced Matlab copy, or to download(/use online? I don't know, which is why you should say) the MRE to make use of the software. Where can they find/download the MRE? Given that the software updates periodically, to comply with code publication best practice please highlight exactly which snapshot of the code this paper specifically describes at time of publication (i.e., as of right now, I would recommend ". . .expanded periodically; this text refers specifically to the code as of commit 8fad562 on 9th July 2018.") Although you say it in the paper, also add for maximum clarity "The TAK manual is also available through Github, or can be downloaded as a supplement of this publication" (or equivalent statement).

0.0.62   Reply:

*Addressed:* We have significantly modified the 'Code Availability' section. We now: 1) specify that the user manual is available on the GitHub page, 2) specify that the versions of the code discussed in this paper refer to the codes as of release v.1.0.2., 3) clarified that the Matlab Runtime Environment is free, and 4) clarify that the GitHub repository includes two versions of the executables (for both Mac OS X and Windows), one that will that install the Matlab Runtime Environment along with the TAK executable and standalone versions of the executables for users that have previously installed the Matlab Runtime Environment. See also our response to Reviewer 3's point number 1.

---

## Author Response (AR2)

*Main Text Comments:*

**P1.L18** – Make 'ksn' italic
**Response:** Change made

**P2.L7** – Make 'ksn' italic here and elsewhere
**Response:** We have changed all instances of ksn in the manuscript to italic

**P4.L2** – 'comamnd'
**Response:** Misspelling of command has been fixed

**P4.L3** – Awkward, rephrase 'avoid or translate from proprietary'
**Response:** This entire sentence was reworded to make the intended statement more clear.

**Figure 2** – Text on figure is too small
**Response:** We have increased the size of text on figure 2

**P8.L7** – 1e9 -> prefer 10^9 format
**Response:** Change made

*Supplement Comments:*

**p.17**: This seems trivial but can you add a citation (or citations) to section 7.2? Is this done using fastscape? Is it d8 flow routing (I'm sure it is but it makes sense to be transparent)?
**Response:** Addressed, detail added. We have added a citation to Schwanghart & Schuler 2014 (which describes the details of how they do flow routing in TopoToolbox, and yes, it does employ a scheme similar to that described by Braun & Willett and implemented in FastScape) for readers interested in the details of the flow routing routines. We have specified that while TopoToolbox can implement multi-neighborhood flow routing algorithms, TAK hard codes in the use of the D8 algorithm because it is simpler (and because using the multi-neighborhood flow routing closes off the usage of some other functions in TopoToolbox).

**Figure 6**: Something has gone wrong on my version: there is a black box in the lower left.
**Response:** This has been repaired.

**Page 31, top**: How is the best fit concavity fit? It sounds algorithmic. But I think this is just a regression through manually picked segments. Is that right? Although the section on page 31 is fairly clear I think it should be highlighted that selection is manual rather than algorithmic. Also, is segment selection reproducible? I'd like to see a few sentences about that. If you select segments for a paper, how is the next author to reproduce your output? Can you read in the segment boundary file?
**Response:** Partially addressed. We have added the requested text on reproducibility and how the concavity fit is done. In regards to how reproducible these results are, and specifically if you could load in someone else's segment boundaries, that's a great point, but in practice because of the way we have written KsnProfiler, it would be a bit of a sticky issue to add this functionality in at present (though we'll definitely put it on the list for future development).

**Page 37**: Can users of the compiled code see any of these outputs? Does it print files in other formats than .mat? It might be useful to reference to the scipy.io module of python, that can read .mat files (https://docs.scipy.org/doc/scipy/reference/tutorial/io.html).
**Response:** Not directly, but we have added some text to instruct users how they can run these functions so that the raster and vector data is output in a readable format (or do it after the fact with the Mat2Arc function) and access the statistics through the use of the CompileBasinStats function. We have also added the suggestion to use the scipy.io module (with a link) to directly read the mat files, thanks for that!

**Top page 41:** Again, I suggest adding a note about the reproducibility of these picked knicks. Can you output a file that allows another user to reproduce the outputs?
**Response:** We added some text here as well. The locations of the picked knicks are output in x,y,z space, but as with segment boundaries in KsnProfiler there is no formal way to load someone else's knickpoints in at present.

**Figure 14:** Is this from the San Gabriel data? Say where it is from. Also, throughout: if the figures are generated from example data that is provided with the package say so.
**Response:** We have specified here that yes, these example data plots are from the Southern California sample dataset, and additionally specified which different basin criteria was used to generate this particular dataset. We have added similar notes to all the figures in the BasinStatsPlots section.

**Figure 15:** These features seem incredibly useful. Nice work.
**Response:** Thanks, we hope so!

**Page 49:** How is the length of the hillslope calculated? It is okay to just refer to the 2016 paper but since this is nontrivial and is essential to interpreting the results I think users should be aware that this is not straightforward at all.
**Response:** The calculation of hillslope length (Lh) is explained on page 50, specifically it is found by (numerically) solving equation 5, which is just finding the length from the divide where the gradient and erosional contributions for the hillslope and the fluvial portion of the landscape are equal given a set of values for threshold gradient, diffusivity, and erosivity.

**Figure 24 caption:** heatmap of what?
**Response:** Have clarified that it just creates a colored map of point density.

**Compare Results**

| Old File: | | New File: |
|---|---|---|
| **TAK_v4.pdf** | versus | **TAK_v5.pdf** |
| **14 pages (20.11 MB)** | | **14 pages (19.79 MB)** |
| 11/24/18, 10:23:45 AM | | 1/1/19, 11:23:45 AM |

**Total Changes**

**161**

**Content**

63 Replacements

31 Insertions

41 Deletions

**Styling and Annotations**

22 Styling

4 Annotations

Go to First Change (page 1)

[revised manuscript text omitted]